# IL-17C is a driver of damaging inflammation during *Neisseria gonorrhoeae* infection of human Fallopian tube

Erin M. Garcia[1,3], Jonathan D. Lenz [1,3], Ryan E. Schaub[1], Kathleen T. Hackett[1], Wilmara Salgado-Pabón[2] & Joseph P. Dillard [1] ✉

The human pathogen *Neisseria gonorrhoeae* ascends into the upper female reproductive tract to cause damaging inflammation within the Fallopian tubes and pelvic inflammatory disease (PID), increasing the risk of infertility and ectopic pregnancy. The loss of ciliated cells from the epithelium is thought to be both a consequence of inflammation and a cause of adverse sequelae. However, the links between infection, inflammation, and ciliated cell extrusion remain unresolved. With the use of ex vivo cultures of human Fallopian tube paired with RNA sequencing we defined the tissue response to gonococcal challenge, identifying cytokine, chemokine, cell adhesion, and apoptosis related transcripts not previously recognized as potentiators of gonococcal PID. Unexpectedly, *IL-17C* was one of the most highly induced genes. Yet, this cytokine has no previous association with gonococcal infection nor pelvic inflammatory disease and thus it was selected for further characterization. We show that human Fallopian tubes express the IL-17C receptor on the epithelial surface and that treatment with purified IL-17C induces pro-inflammatory cytokine secretion in addition to sloughing of the epithelium and generalized tissue damage. These results demonstrate a previously unrecognized but critical role of IL-17C in the damaging inflammation induced by gonococci in a human explant model of PID.

*Neisseria gonorrhoeae* is a Gram-negative pathogen that frequently infects the human reproductive tract. Symptomatic gonorrhea infection is characterized by inflammation at colonized mucosal surfaces and a pronounced influx of neutrophils that are themselves subsequently infected by *N. gonorrhoeae*[1–3]. About two thirds of gonococcal exposures in women result in infection. Yet, an estimated 50–80% of these infections are asymptomatic, resulting in intervention delays[1,2]. Whether initial cervical infection is symptomatic or not, gonococcal ascension from the cervix and infection of upper reproductive tract structures can drive the onset of severe disease sequelae known as pelvic inflammatory disease (PID)[3–5]. Infection of the Fallopian tubes (salpingitis) results in progressive damage (inflammation, ciliostasis,

ciliated cell sloughing, epithelial exfoliation, tubal scarring, and occlusion) that can lead to chronic pelvic pain and an increased risk for infertility and life-threatening ectopic pregnancy[6,7]. The estimated prevalence of PID among sexually active, reproductive-aged women in the United States is 4.1%, with *N. gonorrhoeae* making up a third of all cases[8,9]. This is likely a significant underestimation due to the high rate of asymptomatic presentation and the absence of a diagnostic gold standard[10,11]. Among minoritized populations, PID prevalence may be even more severely underestimated owing to a higher incidence of gonorrhea and poor access to preventative screening and healthcare overall[7,12,13]. Both the rates of reported gonorrhea and the prevalence of PID have increased in recent years[9,14]. This increase is concerning

[1]Department of Medical Microbiology and Immunology, University of Wisconsin-Madison, Madison, WI, USA. [2]Department of Pathobiological Sciences, University of Wisconsin-Madison, Madison, WI, USA. [3]These authors contributed equally: Erin M. Garcia, Jonathan D. Lenz. ✉e-mail: jpdillard@wisc.edu

considering the lack of protective immunity elicited by gonococcal infection, the morbidity associated with PID, and the imminent approach of untreatable gonorrhea[15]. Thus, a better understanding of the mechanisms driving gonococcal PID is necessary to aid the design of effective therapeutic interventions and ultimately reduce the threat it poses to public health.

Natural infection with *N. gonorrhoeae* occurs exclusively in humans due to its exquisite adaptation to the human host[16–18]. This host restriction requires the use of human models to accurately recapitulate in vivo phenomena of gonococcal diseases. The human Fallopian tube organ culture explant (FTOC) model is currently the most physiologically relevant model for understanding human PID[19]. In this model, *N. gonorrhoeae* infection leads to ciliostasis and subsequent sloughing[20–22]. Gonococcal challenge also induces the production of inflammatory cytokines, including TNF-α, IL-1α, IL-1β, IL-8, IL-6, GM-CSF, CCL2 (MCP-1), and CCL4 (MIP-1β)[23–25]. TNF-α levels correlate with loss of ciliary activity in FTOC, and treatment with recombinant TNF-α is sufficient to induce epithelial damage[26,27]. Thus, the host response to gonococci is hypothesized as a major cause of the tissue damage observed during PID, in contrast with the idea that gonococci have a direct toxic effect on host cells. Consistent with this hypothesis, cell-free supernatant or purified gonococcal products, including purified lipooligosaccharide (LOS) or monomeric peptidoglycan (PG), are sufficient to recapitulate epithelial exfoliation characteristic of PID[28–30]. Our current understanding of the pathophysiology of gonococcal PID has been limited to targeted studies in FTOC and many gaps remain. Therefore, *N. gonorrhoeae* elicits a pathological immune response that has not been fully elucidated, yet this knowledge is essential for the development of new intervention strategies.

To address this knowledge gap, we analyzed the transcriptional response (RNA-Seq) of human FTOC to *N. gonorrhoeae* cell free supernatant with the goal of defining the innate response to *N. gonorrhoeae* products that cause ciliated cell damage and death. From this untargeted analysis we observed a significant induction in inflammation and apoptosis-related pathways and identified a multitude of transcripts not previously recognized in association with gonococcal PID. We selected the transcript *IL17C*, a cytokine of the IL-17 family produced specifically in epithelial cells, for further characterization in our model[31]. There has not previously been a recognized role for IL-17C in infections with *N. gonorrhoeae*, nor in any inflammatory conditions of the upper reproductive tract. We found that IL-17C is produced following both cell-free supernatant treatment and *N. gonorrhoeae* infection, and that the IL-17C-specific receptor chain IL-17RE, is present on the Fallopian tube epithelial surface[32]. Recombinant human IL-17C is sufficient to induce cytokines seen during *N. gonorrhoeae* infection, in line with its predicted autocrine, inflammation-amplifying activity[31]. Further, IL-17C in FTOC is sufficient to elicit a pathology hallmark of PID. These results demonstrate that IL-17C is a previously unrecognized component of the host response to gonococci that promotes inflammation and tissue damage within the human Fallopian tube.

## Results

### Inflammatory factors are predominantly upregulated in response to *Neisseria gonorrhoeae* in human FTOC

To understand the response of human Fallopian tube to *N. gonorrhoeae* exposure, we performed RNA-Seq analysis on four donor tissues treated with cell-free supernatant from in vitro growth of *N. gonorrhoeae*. The cell-free supernatants contain factors released and secreted by gonococci including outer-membrane vesicles, lipooligosaccharide, soluble peptidoglycan fragments, heptose-containing metabolites, and a subset of gonococcal proteins either secreted or associated with outer membrane vesicles[28,33–35] (Table S1, Source Data file). Cell-free supernatant was used in place of live gonococci as it recapitulates the major phenotypic features of

infection and allows us to standardize exposures across tissues to avoid the well-known confounding factors of pili and opacity protein variation that accompany *N. gonorrhoeae* infections[21,22,28–30,36]. Tissues were treated for 6 or 24h with supernatant from wild-type *N. gonorrhoeae* strain MS11, and mock-treated, donor-matched tissues were used as controls. At 6h post-treatment, tissue damage is not yet observed in human FTOC, and thus, this timepoint represents the early host response. By 24h, ciliated cell sloughing is apparent. Hence, these timepoints can help distinguish factors important for inducing epithelial damage from those stimulated in response to damage.

The human FTOC transcriptomic response at 6h showed expression of ~15,000 genes in each treatment (experimental or control). Expression levels of 238 of these genes were both significantly different and at a magnitude of 2-fold or greater between treatments (FDR < 0.05, $\log_2 FC \geq 1$ or $\leq -1$) (Fig. 1A). Figure 1B lists the 20 genes most highly upregulated in supernatant treated tissues relative to mock treatment. Seventeen of these genes have known roles in inflammation, including *IL1B*, *NOS2*, and *IL17C*. Many are chemotactic factors that attract a variety of immune cell types (*CCL3*, *CCL4*, *CCL8*, *CCL20*). A complete list of upregulated genes can be found within the Source Data file. Formyl peptide receptor 3 (*FPR3*), a member of the FPR family that functions in the regulation of inflammation, was the only gene repressed more than 2-fold (Fig. 1B)[37].

All significantly upregulated genes listed were entered into three different pathway classification systems to identify biological pathways induced upon *N. gonorrhoeae* treatment of tissues (PANTHER, KEGG, and GO)[38–42]. Common amongst all three analyses was the induction of pathways related to the detection and response to bacterial products, apoptosis, cytokine and chemokine signaling, and inflammation (Fig.1C, D, Fig. S2A). Genes involved in the response to microbial products include receptors known to sense gonococci (*TLR2*, *NOD2*, *NLRP3*), antimicrobial peptides (*DEFB4A* and *S100A9*), and several interferon-induced genes typically associated with viral infection and intracellular bacteria (multiple *OAS*, *GBP*, and *IFI* type genes)[43–45]. Apoptosis pathway genes included both pro- (*TNF*, *LTB*, *GZMB*, *TNFRSF9*) and anti- (*BCL2A1*, *BIRC3*, *NFKB1*, *NKB2*, *NFKBIA*, *TNFRSF6B*, *TNFAIP3*, *TNFAIP8*, *TNFRSF11B*, *PTGS2*, *IFI6*) apoptotic factors. Cytokine and chemokine signaling pathways were comprised of factors previously shown to be induced by *N. gonorrhoeae* during ex vivo Fallopian tube infection (*IL6*, *TNF*, *ILB* and *CCL4* (MIP1β)), as well as many additional inflammatory mediators with heretofore uncharacterized roles in the human FTOC model of gonococcal PID (Source Data)[23,24]. These include CCL and CXCL family cytokines and chemokines that function in the recruitment and activation of neutrophils, monocytes, and T-cells (*CCL2*, *CCL5*, *CCL7*, *CXCL6*, *CCL9*, *CCL10*), as well as factors belonging to interferon gamma (IFN-γ), tumor necrosis alpha (TNF-α), nuclear factor kappa B (NF-κB), and interferon beta (IFN-β) signaling pathways. The interleukin response to supernatant treatment encompassed cytokines with roles in acute inflammation (*IL1B*, *IL1A*, *IL6*, *IL8*, *IL11*) and Th17 polarization (*IL17C*, *IL23A*) in addition to members of the IL-20 family (*IL10*, *IL19*, *IL20*, *IL24*). Cytokines of the IL-20 family facilitate communication between epithelial cells and immune cells and are implicated in multiple inflammatory diseases[46]. In a mouse model of gonococcal infection, IL-10 contributed to suppression of the Th1 response[47]. Pathways involved in IL-17 signaling (23 genes), rheumatoid arthritis (22 genes), cell proliferation (35 genes), and extracellular matrix organization (11 genes) were also enriched.

A similar number of genes were found to be expressed at the 24h timepoint. Of these genes, 61 were both significantly different and expressed with a fold change of 2 or greater between treatments (Fig. S1A). All genes repressed ≥ 2-fold are displayed in Fig. S1B along with the top 20 most highly induced genes. A complete list is included within the Source Data file. In contrast to 6h treatment, inflammatory factors represented less than half of the most upregulated genes, and all but *CSF2* were also upregulated at 6h. Instead, multiple serine protease

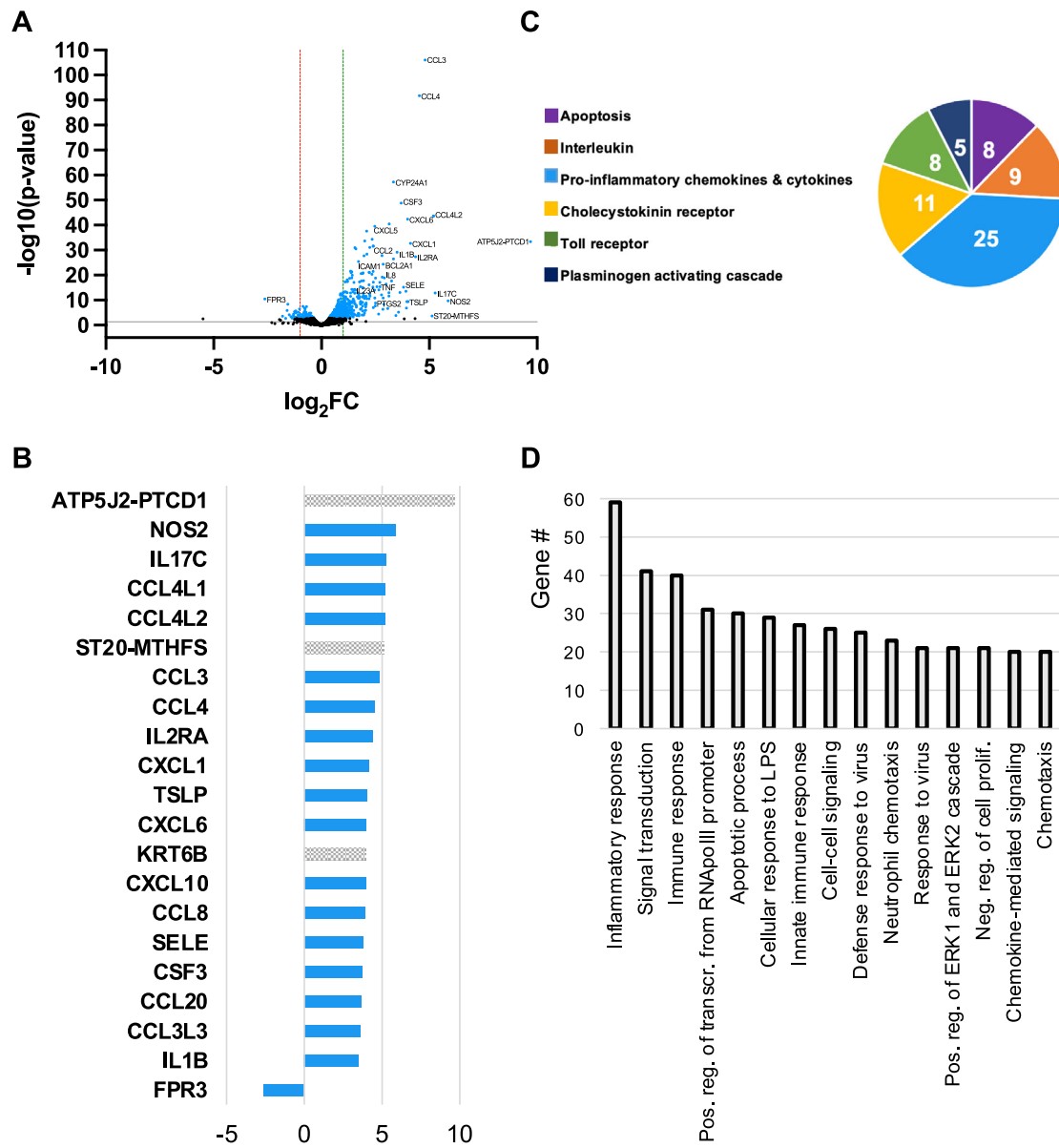

**Fig. 1 | Secreted gonococcal products induce a plethora of pro-inflammatory factors within the human Fallopian tube. A** Volcano plot of all transcripts expressed in mock and 6h *N. gonorrhoeae* supernatant treated tissues. Each dot represents one gene. Those colored black represent genes with a false discovery rate (FDR) > 0.05, while blue dots represent those with an FDR < 0.05. Dots below the gray line represent genes with $p > 0.05$ (-log10(p-value) < 1.3) while those above the line represent those with $p < 0.05$. Genes to the left of the red line are repressed 2-fold or greater ($\log_2$FC ≥ 1). Genes to the right of the green line are induced 2-fold or greater. FC = fold change. **B** Relative expression levels ($\log_2$FC) of genes are significantly different (FDR < 0.05, $p < 0.05$, $\log_2$FC ≥ 1) between treatments. All significantly repressed genes ($\log_2$FC ≤ −1) are displayed. Only the top 20 significantly induced genes ($\log_2$FC ≥ 1) are displayed. The blue bar color is indicative of assignment to a PANTHER, GO, or KEGG inflammation pathway. For A and B, the posterior probability of differential expression and the FDR estimate for multiple comparisons were determined by a Bayesian model (*EBSeq*). **C** PANTHER pathway analysis. The 6 overrepresented pathways are displayed. The number within each slice is the number of significantly different genes assigned to the pathway. D) Gene ontology (GO) biological function pathway analysis, top 15 pathways. Source data are provided in the Source Data file.

inhibitor (*SERPIN*) and *MT1* family genes populate the top 20 upregulated gene list at 24h. *SERPINB3* and *SERPINB4* were also upregulated at 6h, but upregulation of *SERPINB7* and all *MT1* genes was unique to 24h. Downregulation of *FPR3* was maintained at 24h. Additional 24h downregulated genes have functions related to hormone regulation (*SST*, *CHGA*), barrier integrity (*ALPP, CRB2, FLG*) extracellular matrix remodeling (*EGFL7, CILP*), and inflammation control (*CYBB*).

As with 6h, detection and response to bacterial products, cytokine and chemokine signaling, and inflammation pathways were enriched at 24h (Fig.S1C, D, Fig. S2B). However, the inflammatory response was comparatively muted. Pathways involving NF-κB or IFN-β signaling

were no longer enriched, and the interleukin response at 24h was limited to IL-1 and IL-6. Apoptosis pathways were also no longer enriched at this timepoint. Of the apoptotic factors induced at 6h, only *PTGS2* and *IFI6* remained upregulated at 24h. Pathways related to mineral absorption/metal detoxification/zinc homeostasis were uniquely enriched at 24h. The 6 genes classified into these pathways (*MT1M, MT1F, MT1G, MT1H, MT1X, MT1E*) are all members of the metallothionein I family of metal binding proteins that play roles in immune regulation, metal ion homeostasis, and protection against oxidative stress[48–50]. Simultaneous expression of iNOS has been shown to promote the release of intracellular $Zn^{2+}$ from metallothioneins

(MT)[51]. MT induction along with iNOS, *S100A7* (extracellular zinc binding protein) and *SLC30A2* (zinc transport protein) at 24h suggests that zinc regulation may be an important component of the late Fallopian tube response to gonococcal challenge. This protective host response may also be beneficial for the bacteria as *N. gonorrhoeae* is equipped to overcome zinc restriction through piracy of multiple host proteins, including S100A7[52,53].

In sum, results from our RNA-Seq analysis show a strong induction of inflammatory factors in response to gonococcal challenge. These factors represent multiple signaling pathways, highlighting that the Fallopian tube inflammatory response to *N. gonorrhoeae* is more diverse than previously appreciated. That the inflammation signature shares many commonalities with rheumatoid arthritis provides further evidence that gonococcal PID is a disease mediated by the host response to gonococci. While epithelial damage occurs in FTOC in the absence of immune cell recruitment, the cytokines and chemokines upregulated here indicate that an influx of immune cells in vivo could further exacerbate the damaging pathology. Further, recruited adaptive immune cells would encounter a Th17 polarizing environment.

### Gonococcal NOD agonists promote antimicrobial peptide production and repress cell adhesion factors

Because of the demonstrated toxicity of purified monomeric peptidoglycan fragments to Fallopian tube tissues and the abundance of these fragments in gonococcal supernatants, we hypothesized that NOD receptor signaling is a significant contributor to the inflammation that occurs during *N. gonorrhoeae* infection of Fallopian tubes[30,35,54]. Indeed, treatment of human FTOC with purified gonococcal GlcNAc-anhydroMurNAc-L-Ala-D-Glu-*meso*-DAP (TriDAP) causes ciliated cell sloughing, consistent with a role for this NOD1 agonist in Fallopian tube tissue damage (Fig. 2A). Therefore, the RNA-Seq analysis was extended to compare tissues treated with wild type supernatant or supernatant from strain JL539, an L,D-carboxypeptidase A mutant (Δ*ldcA*) deficient in NOD1 and NOD2 activation[55]. The tissues used in this experiment were from the same four donors as above. At both 6 and 24h, a limited number of the ~15,000 expressed genes were induced or repressed at a magnitude of 2-fold or greater. Therefore, the analysis was expanded to include genes with a fold change ≥ 1.5 (log$_2$FC ≥ 0.59 or ≤ −0.59) for both timepoints.

At 6h, *ST20-MTHFS*, a read-through transcript encoding a suppressor of tumorigenicity 20 and 5,10-methenyltetrahydrofolate synthetase fusion protein, was the only gene with a fold change ≥ 1.5 and within significance thresholds (Fig. S3A). At 24h, 2 genes were induced (up in wildtype compared to Δ*ldcA*) and 14 were repressed (Figs. 2B, C). The two induced genes, psoriasin (*S100A7*) and β-defensin 2 (*DEFB4A*), have known roles in antimicrobial defense[56,57]. KEGG analysis classified them both into the IL-17 signaling pathway. The association with this pathway was significant (FDR < 0.05). GO analysis classified a selection of the repressed genes into pathways related to cell-cell adhesion (*LAMA5, CDH5, CD93, PKP1, PCDH12*), tissue development (*LAMA5, HSPG2*), and regulation of extracellular matrix assembly (*NOTCH1, TIE1*), among others (Data S1). However, none of the identified pathway associations were within significance thresholds. Other repressed transcripts excluded from pathway classifications (*LRP1, BMP2, ANPEP*) are also thought to play roles in cell adhesion and matrix remodeling[58–60]. Thus, gonococcal NOD agonists specifically induce mediators of antimicrobial defense while down-regulating factors related to tissue integrity. These results are consistent with previous reports of peptidoglycan-mediated damage in FTOC and highlight the critical contribution of gonococcal NOD agonists to epithelial exfoliation.

### IL-17C is produced in response to *Neisseria gonorrhoeae* in human FTOC

One of the most greatly induced genes ( > 10-fold) following treatment with wild type gonococcal supernatant was *IL17C* (Fig. 1B). IL-17C is a more recently characterized IL-17 family member. It has not previously been implicated in *N. gonorrhoeae* infections, though it was noted to be upregulated in a microarray study of *Chlamydia trachomatis* infection of progesterone-treated endocervical cells[61]. Among the functions ascribed to IL-17C in other systems are autocrine activity that promotes an innate inflammatory response and an ability to potentiate a pro-inflammatory T helper 17 (Th17) cell response[31,62]. Since inflammation is a hallmark of symptomatic *N. gonorrhoeae* infection and inherent to pelvic inflammatory disease, we hypothesized that *IL17C* is involved in the promotion of inflammation by *N. gonorrhoeae*.

To validate the relevance of the discovery that *IL17C* is transcriptionally induced in human FTOC, we treated tissues from 6 additional donors and measured IL-17C protein levels in the tissue medium by ELISA at two different timepoints. For this experiment, infection with a piliated variant of *N. gonorrhoeae* was included as a treatment to ensure IL-17C was not secreted in response to something uniquely produced by bacteria during in vitro growth. Following cell-free supernatant treatment, IL-17C levels were significantly higher at 8h and 24h over mock-treated, paired control samples for each donor (Fig. 3A). Following infection, IL-17C was not yet measurable in all donor samples at 8h but was measurable in all donor samples at 24h (Fig. 3B). Our data confirm that the production of IL-17C in human Fallopian tube tissues is elicited by gonococcal challenge. These results are consistent with RNA-Seq data and affirm that IL-17C stimulating factors produced during in vitro growth are also produced during infection of human FTOC.

Both LOS and monomeric peptidoglycan contribute to tissue damage in FTOC, and many microbial products stimulate IL-17C production in other models[31,63]. Therefore, we questioned whether eliminating stimulatory LOS or NOD1-agonist PG would lower IL-17C responses in FTOC. To that end, RNA-Seq comparisons between wild type and Δ*ldcA* supernatants did not show any differences in *IL17C* expression between treatments. Further, comparisons between wild type and Δ*msbB* supernatants (lacking stimulatory, hexa-acylated lipid A in LOS) in an ELISA assay did not show any differences in IL-17C production ($p = 0.37$). As expected, TNF-α levels were reduced in the Δ*msbB* background ($p = 0.0015$) (Fig. S4). These results indicate that neither microbial product is required for IL-17C production in FTOC.

### IL-17RE is present on the Fallopian tube epithelium

IL-17C is thought to have autocrine, inflammation-amplifying activity in disease models of psoriasis and atopic dermatitis, but in order to serve a similar role in the epithelial inflammation seen in pelvic inflammatory disease, IL-17C would need to be sensed at the epithelial lumen[64]. To determine whether cells in human FTOC are primed for detecting and responding to the release of IL-17C, we performed immunohistochemistry (IHC) for detection of the IL-17C-specific receptor chain. Sensing of IL-17C occurs through a heterodimeric receptor comprised of IL-17RA and IL-17RE, with the IL-17RE chain unique to sensing of IL-17C[32]. Strong and consistent staining was observed for IL-17RE on the simple, columnar epithelial cells lining the lumen, but not within the lamina propria of Fallopian tube samples (Fig. 4). While the distribution of IL-17RE is not well characterized, its localization within the Fallopian tube is consistent with autocrine activity of IL-17C, which is produced predominantly in epithelial cells[65,66]. Thus, human Fallopian tubes can sense the IL-17C produced during infection via receptors present on the epithelial cell surface.

### IL-17C is sufficient to induce pro-inflammatory cytokine secretion from human FTOC

Fallopian tube tissue explants produce IL-17C in response to *N. gonorrhoeae* infection, and the Fallopian tube epithelium contains the IL-17RE receptor chain that participates in the detection of IL-17C. We, therefore, addressed whether IL-17C alone is sufficient to induce an inflammatory response from human FTOC. For this purpose, tissues

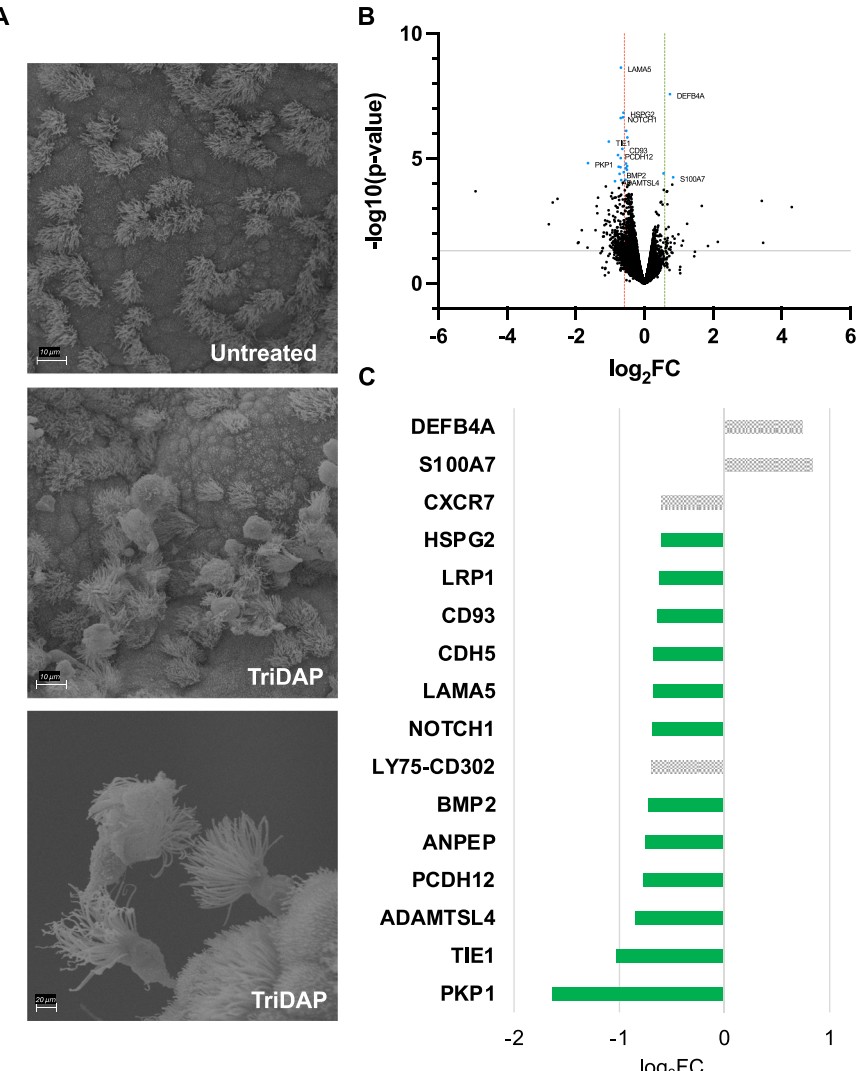

**Fig. 2 | NOD1 agonist promotes damage in FTOC through suppression of genes involved in cell adhesion and extracellular matrix regulation. A** Scanning electron micrographs of control (top) or 25 µg/ml tripeptide monomer (TriDAP) treated (middle and bottom) FTOC. This experiment was repeated independently 5 times with similar results. **B** Volcano plot of all genes expressed in 6h wild type and *ΔldcA* supernatant treated tissues. Each dot represents one gene. Those colored black represent genes with a false discovery rate (FDR) > 0.05 while blue dots represent those with an FDR < 0.05. Dots below the gray line represent genes with *p* > 0.05 (-log10(p-value) < 1.3) while those above the line represent those with p < 0.05. Genes to the left of the red line are repressed 1.5-fold or greater (log₂FC ≥ 0.59). Genes to the right of the green line are induced 1.5-fold or greater. FC = fold change. **C** Relative expression levels (log₂FC) of all genes significantly different (FDR < 0.05, *p* < 0.05, log₂FC ≥ 0.59) between treatments. Green bar color is indicative of a function related to tissue homeostasis (cell adhesion, tissue development, extracellular matrix regulation). For **B** and **C**, the posterior probability of differential expression and the FDR estimate for multiple comparisons were determined by a Bayesian model (*EBSeq*). Source data are provided in the Source Data file.

from three independent donors were treated with 200 ng/mL of recombinant human IL-17C (rhIL-17C) for 6h, and tissues from two donors were treated for 24h. The medium was then assayed for 40 inflammation-related analytes using a semi-quantitative membrane-based antibody array (Fig. S5). As a control, equal portions of each sample were left untreated and for comparison, equal portions were infected with piliated gonococci for 6h and 24h. Based on the array results, IL-8, IL-6, IL-1β, TNF-α, MIP-1β (CCL4), and MCP-1 (CCL2) were then assayed by ELISA from seven additional donor samples. Samples were infected as described for IL-17C quantification or treated with rhIL-17C as above for 6h or 24h.

Infection with *N. gonorrhoeae* produced a significant early[6h] response of IL-1β, TNF-α, and MIP-1β (CCL4) (Fig. 5A). Of those, TNF-α, and MIP1β (CCL4) were also significantly induced by rhIL-17C alone (Fig. 5B). Levels of IL-8, IL-6, and MCP-1 (CCL2) were not yet detectably higher in either infection or IL-17C treatment by 6h (Figs. 5A, B). At 24h,

all the assayed cytokines/chemokines apart from MCP-1 (CCL2) were significantly induced by infection (Fig. S6A). By 24h, IL-17C alone was able to significantly induce IL-8, IL-6, and IL-1β, though it no longer induced MIP1β (CCL4) or TNF-α (Fig. S6B). These results indicate that IL-17C is sufficient to induce an early TNF-α and MIP-1β (CCL4) response, a later IL-8, IL-6, and IL-1β response, and may thus contribute to the pool of cytokines known to be upregulated during gonococcal infection of the genital tract[67]. The presence of MIP-1β (CCL4) as one of the most highly induced transcripts in Fallopian tube explants (Fig. 1B), along with its production from tissue in response to IL-17C, suggests MIP-1β may be a defining feature of inflammation in Fallopian tube.

### IL-17C induces morphological changes in Fallopian tube tissue
Inflammation and epithelial damage within the Fallopian tube is a driver of pelvic inflammatory disease[6]. Gonococcal infection and treatment with gonococcal secreted products, in particular, have

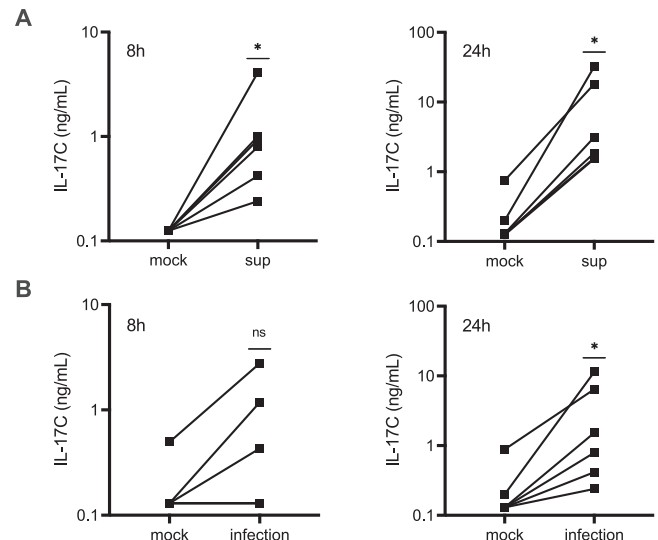

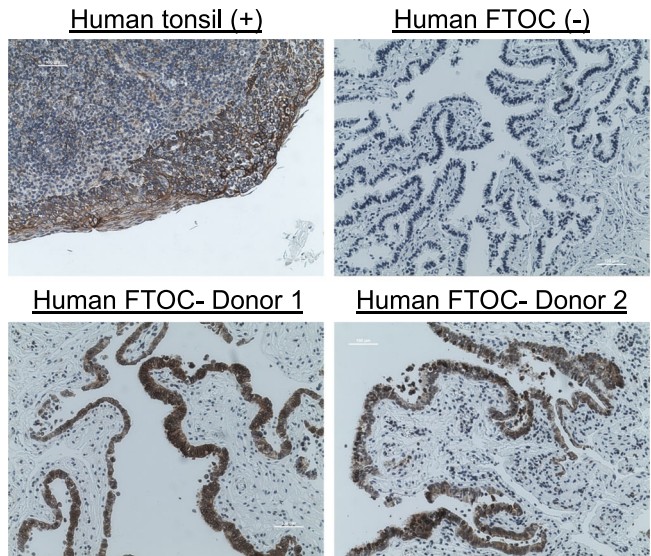

**Fig. 3 | Gonococci induce the secretion of IL-17C in human Fallopian tube tissues.** Quantification of IL-17C levels in mock, wild type gonococcal supernatant, and live gonococci treated tissues via ELISA. A) Comparison of supernatant (sup) and donor-matched mock treated tissues. B) Comparison of *N. gonorrhoeae* infected (infection) and donor matched mock treated tissues. $n = 6$ biologically independent samples. Lines between points represent donor pairs. Each line represents an independent experiment in different donors. Statistical analysis performed using a two-tailed Wilcoxon matched pairs test. * $p < 0.05$, ns = not significant. Source data and exact p-values are provided in the Source Data file.

**Fig. 4 | IL-17RE is expressed in FTOC.** DAB and hematoxylin-stained traverse sections of untreated human Fallopian tube tissues from 2 donors. Negative control sections (-) were stained with secondary but not primary antibodies. Positive control sections (+) from human tonsils were stained with the same concentration of anti-IL-17RE primary antibody as FTOC sections. All images were taken at 10X magnification. This experiment was repeated independently 3 times with similar results.

been shown to induce cell swelling, ciliated cell sloughing, and exfoliation of the Fallopian tube epithelium[19,28–30]. Because IL-17C treatment induced a similar pro-inflammatory response as gonococcal infection in human FTOC, we questioned whether morphological changes to the epithelium would also be similar. Scanning electron microscopy confirmed IL-17C-mediated cell swelling at 24–48h and profuse exfoliation not limited to ciliated cells with prolonged treatment[96h] (Fig. 6). Thus, in addition to inducing inflammation, IL-17C also contributes to epithelial damage in human FTOC. These results provide evidence that IL-17C potentiates the destructive pathology of gonococcal PID.

## Discussion

The molecular mechanisms that drive gonococcal salpingitis and pelvic inflammatory disease (PID) remain poorly defined. Previous studies using human Fallopian tube organ culture (FTOC) have been useful for identifying inflammatory markers and other host factors upregulated in response to gonococci or their secreted products. Yet, they have generally been limited to investigations of specific targets. Many gaps still remain in our understanding of the immune response to gonococci in this context as well as how the immune response facilitates the ciliostasis, ciliated cell death, and epithelial exfoliation characteristic of PID. Therefore, in this study, we characterized the human Fallopian tube response to *N. gonorrhoeae* challenge using an untargeted transcriptomics approach. Focusing on one upregulated transcript from this analysis, we provide evidence that IL-17C is a pro-inflammatory cytokine produced by FTOC in response to gonococcal products. We also provide evidence that IL-17C causes epithelial exfoliation. Our results are consistent with numerous published studies of inflammatory and autoimmune diseases of the skin, joints, intestine, and lungs describing IL-17C as a destructive, inflammation-amplifying cytokine[63,65,66,68–79]. Here, we established a similar role for IL-17C in the context of the reproductive tract. Our work highlights the critical contribution of a novel cytokine to damaging inflammation in a human model of gonococcal PID.

IL-17C has yet to be identified in any context of gonococcal disease or in other models of PID. Previous studies of gonococcal PID using the human FTOC model did not observe IL-17C induction owing to the targeted nature of the study design[23–26]. Even in untargeted studies of gonococcal infection, however, *IL17C* expression was not upregulated[80,81]. These untargeted studies analyzed RNA-Seq data from murine endometrium and human cervix tissues, suggesting that *IL17C* expression in response to *N. gonorrhoeae* may be host-restricted and/or limited to upper reproductive tissues. Human FTOC has also been used to model chlamydial PID[82]. Microarray-based studies of the FTOC response to *Chlamydia trachomatis* infection similarly failed to detect *IL17C* upregulation[82,83]. Thus, IL-17C production during PID could be pathogen specific. TLR2 and NOD2 receptors, which promote IL-17C secretion, are candidates for chlamydial and gonococcal detection in the Fallopian tube[31,84–86]. Ligands for these receptors may be differentially produced between the bacteria during FTOC infection and consequently only promote an IL-17C response to gonococci. Alternatively, it is possible that the time point analyzed in the chlamydial PID model was not early enough to detect *IL17C* induction, as *IL17C* is upregulated early in various disease models[31,32,79]. This possibility is supported by our data showing *IL17C* upregulation at 6h post challenge, but not at the 24h time point. Determining whether IL-17C production is a hallmark of PID and identifying the microbial stimuli promoting IL-17C expression within the urogenital tract are exciting areas for future research. While our data indicate that neither monomeric peptidoglycan nor LOS is required to stimulate IL-17C production in FTOC, gonococci release several other pro-inflammatory products, such as porin (Table S1), outer membrane vesicles, and heptose-containing metabolites, that could serve as candidates for future investigation[33,34,87]

In multiple mucosal infection models, IL-17C signaling plays an important role in enhancing innate epithelial host defenses by stimulating antimicrobial peptide and cytokine expression, maintaining barrier integrity, and promoting tissue repair[74,88,89]. The adaptive immune response is also influenced by IL-17C as Th17 cells express the IL-17C specific receptor and are activated in response to IL-17C

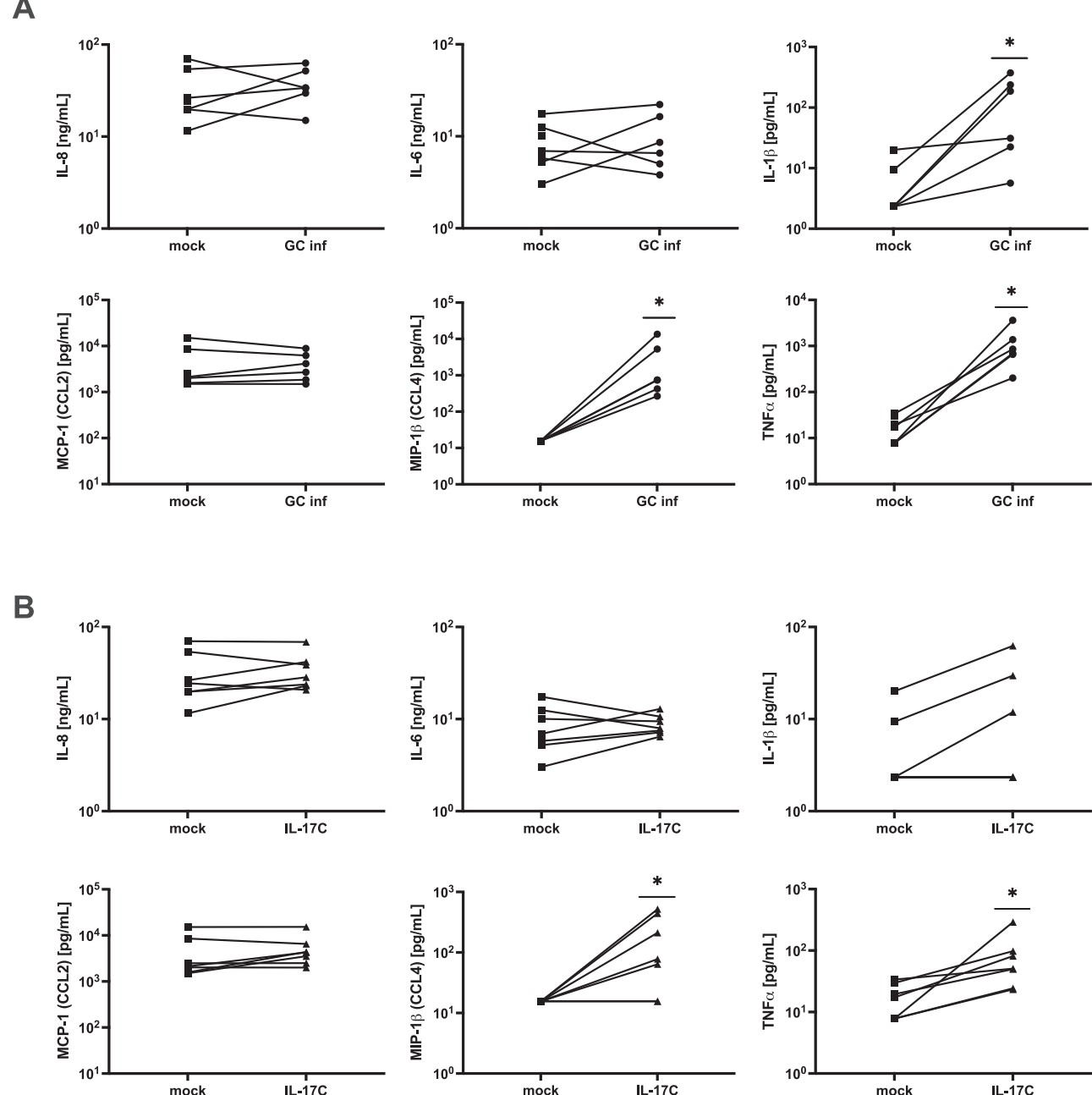

**Fig. 5 | MIP-1β (CCL4) and TNF-α are induced at 6h by both IL-17C treatment and gonococcal infection of FTOC. A** ELISA quantification of 6h mock (square) and *N. gonorrhoeae* infected (circle, GC inf) Fallopian tube tissues. *n* = 6 biologically independent samples. **B** ELISA quantification of mock (square) and 6h 200 ng rhIL-17C (triangle) treated Fallopian tube tissues. *n* = 7 biologically independent samples. Lines between points represent donor pairs. Each line represents an independent experiment in different donors. * *p* < 0.05 Source data and exact p-values are provided in the Source Data file.

engagement[62]. However, in excess, IL-17C production exacerbates inflammation and is a key player in many T cell-dependent and -independent inflammatory conditions[63,65,66,68–79]. Positive feedback from its own autocrine signaling, in addition to synergism with other pro-inflammatory cytokines, greatly amplifies IL-17C-mediated inflammation. Specifically in the context of inflammatory skin diseases, IL-17C potentiates the effects of TNF-α through a pro-inflammatory feedback loop, enhancing its own expression, TNF-α expression, and then synergistically increasing the expression of several other pro-inflammatory cytokines, chemokines, and innate immunity factors[64,65]. Considering the synergistic relationship between IL-17C and TNF-α, our finding that IL-17C treatment induces TNF-α production in human FTOC, and the previously identified contribution of

TNF-α production to tissue destruction in FTOC, we hypothesize that destructive inflammation in gonococcal PID is driven by IL-17C / TNF-α synergy. While adaptive immunity was not characterized in our study, it should be investigated further as a Th17 response to gonococcal infection is thought to contribute to the suppression of lasting protective immunity[90,91]. Whether IL-17C is involved in the Th17 response to *N. gonorrhoeae* has yet to be addressed.

The epithelial exfoliation elicited by IL-17C treatment of human FTOC was not limited to ciliated cells. This contrasts with what is observed in *N. gonorrhoeae* infection or cell-free supernatant treatment of FTOC, where ciliated cells are predominantly extruded. One possible explanation for this result is that gonococcal products inhibit death or exfoliation of non-ciliated cells. In the absence of these

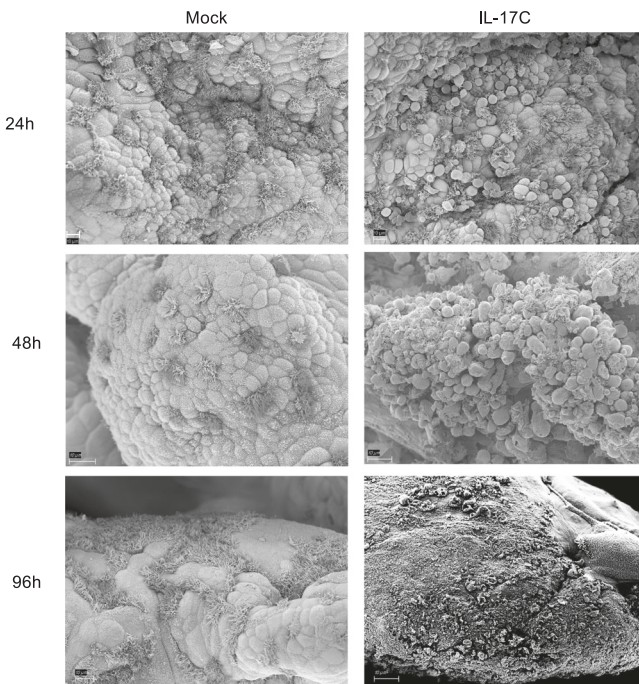

**Fig. 6 | IL-17C treatment promotes epithelial tissue damage in FTOC.** Scanning electron micrographs of mock or 200ng rhIL-17C treated FTOC. Tissues are from three separate donors. This experiment was repeated independently 3 times with similar results.

inhibitory products during IL-17C treatment, the inflammation-induced death response occurs in both ciliated and non-ciliated cells. This hypothesis is supported by studies from multiple groups showing that anti-apoptotic factors, including BCL2A1 (Bfl-1), BIRC3 (c-IAP-2), and PTGS2 (Cox-2), are upregulated upon gonococcal infection of epithelial monolayers lacking ciliated cells[92,93]. Studies of non-ciliated (secretory) Fallopian tube epithelial cells show that gonococcal infection does not induce pro-apoptosis transcripts *TNFRSF1B*, *TNFRSF1A*, *TNFRSF4*, *TNFRSF6*, *TNFRSF10A*, *TNFRSF10B*, *TNFRSF10D*, *BAK1*, *BAX*, *BLK*, *HRK*, or *MCL1* but does induce pro-apoptotic caspase 3[94,95]. Curiously, induction of caspase 3 is only observed at a low multiplicity of infection (MOI). Higher MOIs reduce caspase 3 induction as well as apoptotic DNA fragmentation, putatively due to higher concentrations of an inhibitory soluble factor[95]. These findings from other groups are consistent with our results in human FTOC showing upregulation of anti-apoptosis transcripts, including *BCL2A1*, *BIRC3*, and *PTGS2*, at both 6 and 24h. Further, the lack of caspase 3 induction in our study could be similarly attributable to high concentrations of an inhibitory soluble factor in cell-free supernatants. Apoptosis pathways were enriched in our study at 6h but included both pro-and anti-apoptotic transcripts. At 24h apoptosis was no longer enriched, and the only apoptosis related transcripts upregulated at this time were anti-apoptotic. These results are consistent with the differential fates of secretory and ciliated cells following *N. gonorrhoeae* challenge in human FTOC. Apoptosis may be induced early in ciliated cells to promote their extrusion but inhibited in secretory cells for the duration of gonococcal challenge so that they remain attached.

*Chlamydia trachomatis* is another bacterial pathogen associated with PID[96]. Disruption of epithelial homeostasis within the Fallopian tube is a hallmark of acute *C. trachomatis* infection[82]. This process involves pro-inflammatory cytokine secretion and altered expression of cell adhesion molecules and extracellular matrix (ECM) remodeling enzymes[97]. Our analyzes suggest that gonococcal infection of FTOC also displays these major features, with some notable differences. First, cytokines and chemokines upregulated upon *C. trachomatis*

infection of the Fallopian tube include IL-1α, IL-8, TNF-α, IL-6, LIF, IFN-γ, IFN-β, IL-10, CXCL1, CXCL10, CXCL11, and CCL5[98–103]. Of these, only the interferons were undetected in our study. Although, the GO pathways cellular response to interferon-gamma, positive regulation of interferon-gamma production, and positive regulation of interferon-beta production were enriched. While IFN-β signaling has recently been identified in the context of gonococcal upper genital tract infection, IFN-γ signaling has not[80]. Interestingly, *C. trachomatis* mediated increases in the leukemia inhibitory factor (LIF) cytokine reduced the percent of ciliated cells in Fallopian tube organoids[83]. As functional cilia are essential for oocyte pickup and transport, reducing ciliary activity negatively impacts fertility[104]. Thus, in addition to scarring and occlusion, LIF-induced reduction in ciliated cells may be another mechanism by which gonococcal and chlamydial PID promote tubal infertility.

Inducible nitric oxide synthase (*NOS2*) is upregulated in response to pro-inflammatory cytokines and bacterial stimuli, such as LPS. It catalyzes the synthesis of nitric oxide (NO), which plays a vital role in pathogen defense through direct antimicrobial activity as well as regulation of inflammation and tissue homeostasis[105]. Of note, NO contributes to ciliated cell death in hamster trachea upon exposure to LPS and the monomeric peptidoglycan fragment tracheal cytotoxin (TCT) from *Bordetella pertussis*[106,107]. *NOS2* was one of the most highly induced transcripts in our study at both time points. It is also upregulated during *C. trachomatis* infection of the Fallopian tube[108]. Our result is consistent with a previous report showing *NOS2* induction following gonococcal infection in human FTOC. However, while *NOS2* mRNA was upregulated, nitric oxide synthase activity was reduced, suggesting that levels of NO are reduced upon gonococcal challenge[109]. Reduced NO may promote gonococcal survival within the Fallopian tube. Although, host NO induction occurs in cervix cells following *N. gonorrhoeae* infection and is required for gonococcal survival[110]. NO toxicity is significantly augmented in the presence of oxygen, and cervical oxygen tension is lower than that in the Fallopian tube[111–113]. Thus, whether NO is beneficial or toxic for gonococci may depend on the site of infection. Ultimately, the role of NO in PID and whether it is inhibitory or beneficial to gonococci in this context is uncertain.

Epithelial polarity and adhesion are disrupted during *C. trachomatis* Fallopian tube infection via redistribution and upregulation of adhesion factors[82]. Adhesion molecules upregulated in our study include keratin 6B (*KRT6B*), E-selectin (*SELE*), intercellular adhesion molecule 1 (*ICAM1*), vascular cell adhesion molecule 1 (*VCAM1*), and olfactomedin 4 (*OLFM4*). OLFM4 is a glycoprotein implicated in tumorigenesis within the reproductive tract and is also upregulated by *C. trachomatis* infection[82,114]. Multiple cell adhesion factors were also downregulated in our human FTOC model specifically in response to NOD agonists. It is generally accepted that chlamydial PG is sensed by NOD1 receptors within the reproductive tract[115,116]. Yet, unlike in gonococcal PID, a pathological role for PG in chlamydial PID has not been established. Regarding ECM remodeling enzymes, our data show enhanced transcription of matrix metalloproteases (MMPs) −1, −3, −10, and −12 as well as the metallopeptidase ADAMTS9. NOD agonists promoted downregulation of ADAMTSL4. Other groups have observed MMP-2, −8, and −9 upregulation following *N. gonorrhoeae* challenge of Fallopian tube epithelia[117,118]. In contrast, only MMP-2 and −9 are implicated in *C. trachomatis* mediated Fallopian tube damage. Thus, the epithelium is altered in both gonococcal and chlamydial PID, though likely by different mechanisms. It is unclear from our analyzes whether epithelial homeostasis is disrupted prior to inflammatory damage, as in *C. trachomatis* Fallopian tube infection, or is a consequence of inflammation[82]. These comparisons of gonococcal and chlamydial PID highlight that while major features of pathogenesis are similar, the molecular elements are distinct. This is consistent with clinical data suggesting that in the context of PID *C. trachomatis*

infection is more frequently asymptomatic while *N. gonorrhoeae* infection is more severe[119]. Whether IL-17C contributes to these differences is a topic for future study.

Collectively, this work demonstrates that the transcriptional response of the human Fallopian tube to gonococcal secreted products is characterized by robust inflammation involving multiple inflammatory pathways and changes to apoptosis-related signaling networks and epithelial homeostasis. Further molecular characterization will be required to ascertain the causality of identified transcripts in specific phenomena in the context of the Fallopian tube (suppression of apoptosis, epithelial remodeling, zinc homeostasis, etc.) and to delineate the signaling networks connecting bacterial stimuli to damaging inflammation. The data further show that the cytokine IL-17C has a pro-inflammatory role previously unappreciated in *N. gonorrhoeae* infection. Its presence in a human tissue model of PID suggests it contributes to natural disease in humans. The focus of this study was salpingitis-mediated PID, but infections of any upper reproductive tract tissue can cause PID. While the IL-17C-specific receptor seems to be expressed throughout the reproductive tract, it is currently unknown if other reproductive tissues produce IL-17C in response to gonococci[120,121]. Continued research on the pathophysiology of PID, and whether IL-17C may be a common feature, is crucial because by the time infection is detected and treated, the epithelium is often already irreversibly damaged. In fact, approximately half of women with PID are asymptomatic yet still experience adverse reproductive outcomes[122,123]. Further characterization of the early host response to infection, in particular, may identify biomarkers of PID that could improve screening and accelerate intervention. Finally, the role of gonococcal induced IL-17C in the development of the Th17 response should be investigated further as the inability to develop lasting protective immunity against *N. gonorrhoeae* infection remains a major public health challenge, especially in the face of increasing antibiotic resistance[124].

## Methods

### Ethics statement
This research complies with all relevant ethical regulations. Use of de-identified human tissue for these studies was approved by the Institutional Review Board of the University of Wisconsin-Madison, Protocol #2014-0874, and determined to be exempt as human subjects research. Human Fallopian tube samples were obtained from consenting donors via the National Human Tissue Resource Center at the National Disease Research Interchange (NDRI) in accordance with the University of Wisconsin-Madison Protocol #2014-0874. Donors were aged 50 or younger and undergoing elective hysterectomy or salpingectomy for non-cancer indications unrelated to pelvic inflammatory disease.

### Gonococcal strains
*N. gonorrhoeae* MS11 was used as the wild-type strain in this study, and both mutants (Δ*ldcA* and Δ*msbB*) were derived from this strain. Strain JL539 Δ*ldcA* was generated as previously reported[55]. Strain Δ*msbB* was generated by interrupting *msbB* with a kanamycin resistance cassette[125]. *N. gonorrhoeae* was grown at 37 °C + 5% CO$_2$ on gonococcal base (GCB) agar plates (Difco) containing Kellogg's supplements[126,127]. Infections of human tissue were initiated from overnight agar plates of piliated, Opa- gonococci.

### Cell-free supernatant
For preparation of cell-free supernatant, overnight plates of non-piliated bacteria were used to initiate liquid cultures at OD$_{540}$ 0.25 in gonococcal base liquid medium (GCBL) containing Kellogg's supplements and 0.042% NaHCO$_3$ (complete GCBL or cGCBL)[128]. Cultures were incubated at 37 °C with rotation for 3h, the bacteria collected by centrifugation, and the supernatant filtered through a 0.2 μm PES

filter. Supernatants were normalized based on total protein of the cell pellet, quantified by bicinchoninic acid (BCA) assay (Pierce). Methods for mass spectrometry assessment of proteins in cell-free supernatant are detailed in the Supplementary Information document.

### Tissue processing and treatment for RNA-Seq
Tissues were shipped overnight on wet ice in Dulbecco's Modified Eagles Medium (DMEM) with penicillin and streptomycin. Upon arrival, human Fallopian tube samples were processed immediately and placed in Eagles Minimal Essential Medium (MEM) (Cellgro) buffered with 1M HEPES to pH 7.4 and penicillin-streptomycin. For all experiments, connective tissue, sample ends, and outer muscle layers were trimmed to remove extraneous material around the Fallopian tube. For samples used in RNA-Seq experiments, tissues were cut longitudinally to expose the epithelia of the lumen, and equal amounts of tissue were divided between treated and untreated conditions in 3 ml media volumes in a 12-well tissue culture plate. Samples were allowed to equilibrate overnight following dissection, then washed twice with sterile PBS and placed in fresh medium. For treatment of tissues, *N. gonorrhoeae* was grown in liquid culture (as noted above) for 2.5 h, at which time, bacterial cells were harvested by centrifugation, and conditioned medium was filtered through a 0.2 μm polyethersulfone (PES) filter. Bacteria were subject to detergent lysis, and total cellular protein was measured by bicinchoninic acid (BCA) assay (Pierce). Prior to tissue treatment, conditioned media were normalized using the total cellular protein measurement to standardize experiment-to-experiment differences in bacterial growth. Treated wells received 20 μl/ml of filtered, normalized gonococcal-conditioned media, and mock-treated wells received an equal volume of culture medium (cGCBL). Following a 6 or 24 h incubation, tissues were removed from media for RNA extraction/preservation. The above treatment was repeated with four separate human donors.

### RNA extraction
At the conclusion of treatment, samples were either processed immediately or placed in RNA*later* (Ambion) at 4 °C overnight, followed by storage at −80 °C. Prior to homogenization, RNA*later* was removed (if necessary), and each sample was placed in 0.5 ml Qiagen RLT buffer + 5 μl 2-mercaptoethanol in a bullet-bottom tube with 1 x 5 mm RNase-free stainless steel bead. Homogenization was performed at 30 Hz for 4 m. Samples were then incubated at 37 °C for 30 m with the addition of 1 ml of 200 μg/ml Proteinase K (Sigma-Aldrich). The total contents of each digest were then applied to a Qiashredder column (Qiagen) and harvested by centrifugation at 10,000 x *g* for 2 m. Flow-through was transferred to a new 1.5 ml tube and undigested debris removed by centrifugation at 10,000 x *g* for 3 m, followed by transfer of cleared lysate to a new 1.5 ml tube. To precipitate RNA, one volume of 70% ethanol was added to each sample and mixed thoroughly prior to transfer to a Qiagen RNeasy mini spin column. RNA was washed and eluted according to the manufacturer's recommendations for the RNeasy Mini Kit (Qiagen). Following elution, samples were DNase treated with the TURBO DNA-*free*™ Kit (Ambion) according to the manufacturer's instructions, then precipitated and purified again with the RNeasy Mini Kit, as above.

### Library preparation and RNA sequencing
Purified RNA was submitted to the UW Biotechnology Center's Gene Expression core facility. Quality analysis was performed with an Agilent RNA PicoChip. Ribosomal RNA was removed using the Ribo-Zero rRNA reduction mix (Illumina), and cleanup was performed with RNAClean XP beads (Beckman Coulter). Equal amounts of RNA from technical replicates within each treatment condition (processed separately) were then pooled to generate one library per condition, per sample. Stranded cDNA libraries were generated using the TruSeq Stranded Total RNA Library Prep Kit (Illumina). Prior to sequencing, cDNA

libraries were purified using Agencourt AMPure XP beads (Beckman Coulter), and 3' ends were adenylated, followed by adapter ligation and two additional AMPure XP purifications. Twelve cycles of PCR were performed to enrich for adapter-containing fragments, followed by an additional AMPure XP purification and sequencing on a HiSeq 2000 (Illumina) to generate single-end 100 bp reads.

## Bioinformatics analysis

Raw data from RNA-Seq was analyzed by the UW Biotechnology Center's Bioinformatics Resource Center. Sequencing reads were trimmed to remove sequencing adapters and low quality bases using Skewer, then aligned to the annotated human reference genome using the alignment tool STAR[129,130]. Quantification of expression for each gene was calculated by RSEM, which produces both Transcripts Per Million reads (TPM) and expected read count[131]. The expected read counts were used to determine differential gene expression using EBSeq[132]. Average fold change was calculated for the set of four independent patient samples (average mock vs average *N. gonorrhoeae* treated, or average wild type treated vs average *ΔldcA* treated). To be considered as differentially expressed, transcripts had to be $> 0.59 \log_2FC$ (1.5-fold) different for comparisons of wild type vs *ΔldcA* or $> 1 \log_2FC$ (2-fold) different for comparisons of mock vs *N. gonorrhoeae*, with $P < 0.05$ and a false discovery rate (FDR) $< 0.05$.

## Pathway analysis

Lists of differentially expressed genes were submitted to Protein Analysis Through Evolutionary Relationships (PANTHER) (version 17.0) both for functional classification and to test for statistical over-representation of PANTHER pathways. A Fisher's Exact test with False Discovery Rate (FDR) correction was used to test for over-representation. P-values $< 0.05$ were considered significantly over-represented. Lists were also submitted to DAVID Bioinformatics Database (version 2021) for both functional classification and enrichment analysis using Kyoto Encyclopedia of Genes and Genomes (KEGG) and Gene Ontology (GO) (Biological Process, Direct) pathways. P-values were FDR corrected and considered significant at values below 0.05. For the 6h WT supernatant vs mock treatment gene list, PANTHER pathway analysis sorted the selected genes into 35 different pathways (Source Data file). Those overrepresented (FDR $< 0.05$) are presented in Fig. 1C, and their gene assignments are listed in the Source Data file. Functional enrichment analysis performed with GO revealed 232 biological pathways significantly associated with the gene list. The 15 pathways with the greatest number of gene assignments are detailed in Fig. 1D while all pathways and their gene assignments are listed in the Source Data file. KEGG analysis identified 50 pathways significantly associated with the gene list. The 15 pathways with the greatest number of gene assignments are included in Fig. S2A, and the remaining pathways identified can be found in the Source Data file. For the 24h WT supernatant vs mock treatment gene list, pathway analysis was performed on both repressed and induced genes, though no pathway associations with the repressed gene list were within significance thresholds. PANTHER analysis of 24h induced genes categorized them into 9 different pathways, but only two pathways were assigned more than one gene (Fig. S1C). Significant associations were made with 26 GO and 20 KEGG pathways (top 15 in Figs. S1D and S2B, respectively, with the complete lists included in the Source Data file).

## Tissue harvest, fixation, and preparation for histology

For histological analysis, donor tissue was received and processed as above but instead cut as transverse sections 0.5–1 cm in length. Tissues were fixed in a 4% MeOH-free formaldehyde solution prepared in PBS for 2–4 days at 4 °C, then moved into 70% EtOH. Tissues were oriented during paraffin embedding to produce cross-sections of the lumen when sliced and mounted on glass slides.

## Immunohistochemistry

Paraffin-embedded cross-sections of tissue were heated in a 60 °C oven for 20 m, followed by a paraffin removal and re-hydration series of xylene (3 x 3 m), 100% ethanol (2 x 2 m), 95% ethanol (2 x 2 m), 70% ethanol (2 m), and distilled $H_2O$ (d$H_2O$). Antigen retrieval (HIER) was performed in citrate buffer pH 6.0, in an 80 °C $H_2O$ bath for 2.5 h, followed by a 20 m incubation at room temperature, and d$H_2O$ rinse. Samples were permeabilized for 10 m with 0.5% Triton X-100 in PBS, followed by 2 x 5 m PBS washes. Endogenous peroxidase activity was blocked by a 10 m treatment with 3% $H_2O_2$ in PBS, followed by 2 x 5 m PBS washes. Non-specific background staining was blocked by a 60 m treatment with 10% BSA. Sections of human Fallopian tube were incubated with a 1:200 dilution of antibody directed against IL-17RE (NBP1-93925, Novus Bio, manufacturer validated) at 4 °C overnight, followed by 2 x 5 m TBS washes. Samples were then incubated with horse anti-rabbit IgG-HRP antibodies (MP-7401, Vector Laboratories) for 35 m at room temp, followed by 2 x 5 m TBS washes. Positive control tissues (normal human tonsil) were stained in parallel and additional negative control slides of Fallopian tube were prepared with only secondary (not primary) antibody incubation. All samples were incubated with the chromagen 3,3'-diaminobenzidine (DAB) for 2 m, washed for 5 m in d$H_2O$, and counterstained in Harris Hematoxylin for 2 m, followed by a d$H_2O$ rinse. Stained slides were immersed in 1% acid alcohol, rinsed with d$H_2O$, immersed for 30 s in 1% ammonia solution, and rinsed again with d$H_2O$. Samples were dehydrated in 95% ethanol (10 immersions in 2 baths), 100% ethanol (10 immersions in 2 baths), and xylene (10 immersions in 3 baths). Coverslips were then applied and sealed with Cytoseal mounting medium (Richard-Allan Scientific). Images were obtained on a Nikon Eclipse TiS/L100 inverted fluorescent microscope equipped with a Lumenera Infinity color camera, using the Nikon NIS Elements software. All images were subjected to identical brightness and contrast adjustments then cropped to publication size in Adobe Photoshop CS5.

## Tissue processing and treatment for cytokine measurement

For membrane array and ELISA analysis of cytokine induction following infection or rhIL-17C treatment, donor tissue was received and processed as above for RNA-Seq, with equal amounts of each donor tissue placed in triplicate wells for exposure to either 1) 200 ng/ml rhIL-17C, 2) 20 μl/ml of piliated *N. gonorrhoeae* at 1.0 $A_{540}$ equivalent suspended in 1 ml cGCBL (average inoculum of $2.1x10^7$ CFU/ml), or 3) cGCBL medium only (mock). Medium at the time of exposure was DMEM buffered with HEPES to pH 7.4 and streptomycin (no penicillin). *N. gonorrhoeae* strain MS11 is naturally streptomycin resistant. Following exposures above, media samples were harvested at 6 and 24 h, filtered through a 0.2 μm PES filter, and stored at −80 °C. The above treatment was repeated with a total of seven donor samples independent from those used for RNA-Seq.

For analysis of IL-17C induction by gonococcal infection or supernatant, donor tissue was processed similarly to other experiments, except that following longitudinal opening of the tube, sections were created with a 3 mm biopsy punch. Five punches were added per 1 ml of tissue culture medium in wells of a 24 well plate. For donor tissues treated with gonococcal cell-free supernatant, bacteria were grown and supernatant prepared as detailed above. For donor tissues infected with *N. gonorrhoeae*, piliated gonococci were grown and inoculated as above but following normalization to 0.1 $A_{540}$ equivalent in 1 ml cGCBL (average inoculum of $5.3x10^6$ CFU/ml). For all tissues in both experiment types, addition of cGCBL medium to tissue was used as a mock-treated control. Following exposure to supernatant or infection, media samples were harvested at 8 and 24 h, filtered through a 0.2 μm PES filter, and stored at −80 °C. Supernatant and infection treatments each utilized six different donor tissues, all independent from those used for other experiments.

## Cytokine analysis

Human inflammation antibody array membranes (Abcam) were used as an initial screen for measurable cytokines induced by rhIL-17C or *N. gonorrhoeae* infection from four donor samples of the eventual total of seven donor samples noted above. Twelve of the 40-analyte membranes were blocked with the supplied assay buffer for 30 min at room temperature prior to the addition of 1 ml/membrane of the harvested media from mock, rIL-17C-treated, or *N. gonorrhoeae*-infected donor samples. Membranes were incubated with samples at 4 °C overnight, washed, and incubated with a biotin-labeled anti-cytokine mix at 4 °C overnight, followed by another wash series and incubation with HRP-conjugated streptavidin for 2 h prior to a final wash. Chemiluminescent detection was performed according to the manufacturers' instructions and membranes were imaged on a LiCor Odyssey imager. Densitometry data was captured using the Grid Array Analysis function in LiCor Image Studio and exported to Microsoft Excel for analysis. Positive control spots on each array were used to normalize between samples (with the mock array set as a reference).

Quantitative measurement of individual cytokines was performed by sandwich ELISA using Invitrogen uncoated ELISA kits for IL-8, IL-6, IL-1β, and TNF-α; and R&D Systems DuoSet ELISA kits (with DuoSet Ancillary Reagent Kit 2) for MIP-1β (CCL4), MCP-1 (CCL2), and IL-17C, according to the manufacturer's specifications. IL-17C assays included in Supplementary Fig. 4 were performed using the Human IL-17C ELISA Kit (Abcam). For all assays, 100 μl of the sample was tested in duplicate wells. Plates were read on a Synergy HT microplate reader (BioTek) at $A_{450}$ with background subtraction at $A_{570}$. For all cytokine ELISAs, a 20% standard deviation cutoff was used for inclusion.

## Purification of tripeptide monomer

GlcNAc-anhydroMurNAc-L-Ala-D-Glu-*meso*-DAP (Tripeptide monomer or TriDAP) for treatment of human FTOC was purified from whole gonococcal sacculi exactly as previously described[133]. Sacculi were extracted from cells by boiling in 4% SDS for ~1h and then digested stepwise with 1.4 μM purified NGO1686 (37 °C, overnight) and 0.5 μM purified LtgA (37 °C, overnight). The digested material was then boiled, centrifuged, and filtered through a 10,000 MW cutoff filter to remove insoluble material and large molecular weight molecules. Monomeric peptidoglycan fragments were separated by reversed-phase HPLC, using a Waters Xselect CSH C18 preparative column (5 μm, 10 x 250 mm) and a 4–13% acetonitrile gradient over 30 min. Fractions containing the desired monomer (tripeptide) were pooled, lyophilized, and suspended in water. Quantification was performed with the Fluoraldehyde o-Phthaldialdehyde (OPA) reagent (Thermo Scientific Pierce).

## Tissue treatment and processing for scanning electron microscopy

Tissues were processed similarly to other experiments with 3–7 mm sections cut with a biopsy punch following longitudinal opening of the tube. Tripeptide monomer was added to tissues at 25 μg/ml for 24 h, and rhIL-17C was added at 200ng/ml for 24–96 h[31]. Following treatments, tissues were rinsed in PBS and fixed overnight in 3% glutaraldehyde (pH 7.3 in 0.2M sodium cacodylate) at 4 °C. Tissues were washed 2X with 0.2M sodium cacodylate for 10 min each and post fixed with 2% osmium tetroxide prepared in 0.2M sodium cacodylate for 30 min. After 2X ddH$_2$O rinses for 10 min each, tissues were dehydrated in the following ethanol series for 10 min at each concentration: 30%, 50%, 70%, 80%, 90%, 95%, 3 X 100%. Tissues were then critical point dried with CO$_2$ for 3 exchanges, mounted onto aluminum specimen mounts, and sputter coated with 14nm of platinum coating. Specimens were stored in a desiccator until imaging. Imaging was performed on a Zeiss LEO 1530 scanning electron microscope at 3 kv.

## Statistical analysis

For all statistical analysis of cytokine data, GraphPad Prism 4.0 was used, with significance levels noted in the appropriate figure legend. To determine the appropriate statistical test to use for the analysis of cytokine assays, data were first tested for normality by the Shapiro-Wilk normality test. Normally distributed data were analyzed by paired *t*-test, and data not normally distributed were analyzed using the Wilcoxon matched pairs test. Information on the specific test used and exact p-values are detailed in the Source Data file for each figure.

## Reporting summary

Further information on research design is available in the Nature Portfolio Reporting Summary linked to this article.

## Data availability

RNA sequencing data supporting this study's findings have been deposited in the Gene Expression Omnibus (GEO) database under the accession identifier GSE253499. The uploaded data include raw counts as fastq files for each sample and processed counts as transcripts per million (TPM). The mass spectrometry data generated in this study have been deposited to the ProteomeXchange Consortium database via the PRIDE partner repository under accession code [http://www.ebi.ac.uk/pride/archive/projects/PXD051157]. The following databases were used for data analysis: DAVID Bioinformatics Database, version 2021 [https://david.ncifcrf.gov/home.jsp]; PANTHER classification system, version 17.0 [http://www.pantherdb.org]; Mascot search engine, in-house v2.7.0. Source data are provided with this paper.

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

## Acknowledgements

This work was supported by NIH grants R01AI097157 (J.P.D.), F32AI115911 (J.D.L.), and T32AI055397 (J.D.L. and E.M.G.). We would like to thank

Sandra Splinter BonDurant, Marie Adams, and Anne Luebke at the UWBC Gene Expression Center for RNA-Seq support; Xiaoyu Liu at the UWBC Bioinformatics Resource Center for RNA-Seq Analysis; and Derek Pavelec for bioinformatics consultation. We thank Richard Knoll of the Materials Science Center for training in electron microscopy. We thank Satoshi Kinoshita of the Translational Initiatives in Pathology Laboratory, UW Department of Pathology and Laboratory Medicine for tissue processing and slide mounting, and Sierra Raglin of the UW Department of Surgery Histology Core for immunohistochemical staining. We thank the University of Wisconsin Biotechnology Center Mass Spectrometry Core Facility for the mass spectrometry analysis of membrane-associated proteins in the cell-free supernatant.

## Author contributions

J.D.L. and J.P.D. designed the project. J.D.L., E.M.G., R.E.S, and K.T.H. performed experiments. E.M.G, R.E.S., and J.D.L. performed data analysis and data visualization. E.M.G., J.D.L., and W.S.P. wrote the manuscript. J.P.D. and W.S.P. assisted with manuscript editing.

## Competing interests

The authors declare no competing interests.
