## [Peer Review File · Nature Communications]

IL-17C is a driver of damaging inflammation during *Neisseria gonorrhoeae* infection of human Fallopian tubeREVIEWER COMMENTS

Reviewer #2 (Remarks to the Author):

This work describes the role of IL-17C on ex vivo cultures of human Fallopian tubes after infection with *Neisseria gonorrhoeae* (Ngo) or "cell-free supernatant" treatment.

The MS has significance from a descriptive point of view of the role of IL-17C mediating damage of cultures of human Fallopian tubes. It complements other findings on the estradiol-treated female mice as a surrogate host for Ngo genital tract infections and serum levels from the infected human being.

Data interpretation and conclusions are based on comparing infection with Ngo and cell-free supernatant treatment of Fallopian tubes, considering only LOS and PG as elements that cause ciliated cell damage and death. In this line, the results corresponding to gonococcal NOD agonists seem out of context from the other results and require complementing using a LOS mutant of Ngo to address this point. Moreover, it is unclear how authors define "the cell-free supernatant"; there is no characterization of all components of bacteria supernatant. Also, a feature of Ngo is to produce OMV, and it was demonstrated that OMV resembles the bacteria and has a significant effect on target cells in an inflammatory context. Include experiments with OMV or have the cell-free supernatant characterization improve the robustness of this work. Authors also try to avoid other virulence factors such as Pili and Opa; I think a good rationale for this is to include Pili mutants. On the other hand, authors showed the participation of IL-17C by RNA seq using rhIL-17C and its receptor by IHQ. To have complete confirmation, blocking the IL-17C produced by explants after the challenge is necessary. Regarding cytokine production after Ngo infection and supernatant treatment, the membrane array did not show a difference between the conditions, but ELISA analysis of cytokine induction does. It is essential to address this point in the discussion section.

Data are well described. However, in the methodology concerning RNAseq, it is essential to indicate which bioinformatics tools were used to model the results and make data comparisons. I mean, what statistical model was used to compare significantly different genes?

Reviewer #3 (Remarks to the Author):

Using an unbiased approach, Garcia et al show that IL-17C drives inflammation mediated by *N. gonorrhoeae* or its culture supernatants in fallopian tube organ culture (FTOC) explants. They show expression of the IL-17C receptor chain, IL-17RE, on epithelial cells. They postulate that autocrine signaling by IL-17 through IL-17-RE mediates inflammation and fallopian tube damage. Inflammation could also be incited by peptidoglycan tripeptide monomers and also by treatment of fallopian tube explants with recombinant IL-17C. Detailed pathway analyses have been provided. The data are novel, clearly presented and support the conclusions. A couple of minor points need clarification.

1. Estrogen upregulates IL-17 expression by T cells. Is there any evidence that IL-17C or IL-17RE are estrogen (or progesterone) regulated? This would be an important consideration for inflammatory effects in the female genital tract.

2. Line 315 – I presume this is 200 ng/mL? How was this concentration selected? It seems to be much higher than the IL-17C levels detected in supernatants (eg, Fig. 3).

REVIEWER COMMENTS

Reviewer #2 (Remarks to the Author):

This work describes the role of IL-17C on ex vivo cultures of human Fallopian tubes after infection with *Neisseria gonorrhoeae* (Ngo) or “cell-free supernatant” treatment.

The MS has significance from a descriptive point of view of the role of IL-17C mediating damage of cultures of human Fallopian tubes. It complements other findings on the estradiol-treated female mice as a surrogate host for Ngo genital tract infections and serum levels from the infected human being.

Indeed, nice work from Feinen B et al. (2010. *Mucosal Immunol.*) uncovered the role Th17 cells in the gonococcal mouse model, and Francis IP et al. 2018 (*BMC Genomics*) described the transcriptomic response in mouse upper genital tract to gonococcal infections, which we discussed. Interestingly, IL-17C was not made by the mice, suggesting the response we have studied here may be human-specific.

Data interpretation and conclusions are based on comparing infection with Ngo and cell-free supernatant treatment of Fallopian tubes, considering only LOS and PG as elements that cause ciliated cell damage and death. In this line, the results corresponding to gonococcal NOD agonists seem out of context from the other results and require complementing using a LOS mutant of Ngo to address this point. Moreover, it is unclear how authors define “the cell-free supernatant”; there is no characterization of all components of bacteria supernatant. Also, a feature of Ngo is to produce OMV, and it was demonstrated that OMV resembles the bacteria and has a significant effect on target cells in an inflammatory context. Include experiments with OMV or have the cell-free supernatant characterization improve the robustness of this work. Authors also try to avoid other virulence factors such as Pili and Opa; I think a good rationale for this is to include Pili mutants.

We have included additional data and performed additional experiments to address the reviewer’s concerns. The cell-free supernatants are made by simply centrifuging the gonococcal culture and passing the supernatant through a 0.2 μ filter, the same way they were made in the original experiments from Zell McGee’s studies showing that the supernatants caused damage to Fallopian tube tissue (Melly et al. 1981 *JID* 143:423), the same way we make supernatants for analyzing peptidoglycan fragments released by gonococci (Chan JM and Dillard JP 2017. *J. Bacteriol.* 199:e00354-17 for a review), and the same way that supernatants were made that showed the effects of heptose metabolites on gonococcal infection (Malott et al. 2013. *PNAS* 110:10234). See Materials and Methods lines 567-573. Thus the cell-free supernatants contain the molecules smaller than 0.2 μ that are released or secreted by gonococci.

As the reviewer points out, gonococci make lots of OMVs, and they have inflammatory effects. Since the OMVs are substantially smaller than 0.2 μ , OMVs are abundantly present in cell-free supernatants. We characterized the cell-free supernatants by harvesting the OMVs from the supernatants and subjecting them to mass spectrometry. We found these contained a

variety of outer-membrane proteins as would be expected, including the porin, a known stimulant of immune responses in gonococcal infection (Massari P et al. Trends Microbiol. 2:87-93). We added these new data to the manuscript as Table S1.

We also performed experiments to determine if LOS is a significant contributor to the IL-17C response we studied in this manuscript. We used an *msbB* mutation we got from Mike Apicella. This mutation causes the bacteria to make penta-acylated LOS instead of hexa-acylated, and the LOS is not stimulatory via TLR4 (Post DMB et al. 2002. Infect Immun 70:909). We measured IL-17C and TNF-alpha from FTOC following treatment with WT or *msbB* supernatant. The *msbB* mutant was not different from WT for inducing IL-17C production, while TNF-alpha levels were reduced in the *msbB* experiment, as would be expected due to the change in LOS structure. These results demonstrate that LOS is not required for the IL-17C response. The new data are in Fig. S4.

As for performing infections with strains locked for pilin or Opa expression, that would be a great idea for future experiments. For the present manuscript, infection with gonococci resulted in the same responses as supernatant addition (Fig. 3, Fig. S5), negating the need for additional experiments with *opa* or pilin mutants here.

On the other hand, authors showed the participation of IL-17C by RNA seq using rhIL-17C and its receptor by IHQ. To have complete confirmation, blocking the IL-17C produced by explants after the challenge is necessary.

We fail to understand the importance of this experiment. We show in Fig. 3 that gonococcal supernatant treatment or gonococcal infection of FTOC results in IL-17C production as predicted by the RNA-Seq results. We show in Fig. 5 that recombinant human IL-17C induces the same inflammatory cytokines as gonococcal infections (TNF-alpha, MIP-1beta, and IL-1beta) and not the others (IL-8, IL-6, and MCP-1). We show in Fig. 4 that the IL-17C specific receptor (IL-17RE) is readily detectable on the Fallopian tube epithelial cells. We don't need to go back and demonstrate that IL-17C exclusively binds IL-17RE, as that was already done by others (Chang SH et al. Immunity 35:611-21). Since we can see the effects by adding only purified IL-17C, then we think we are safe in concluding that the results are due to IL-17C binding to its receptor.

Regarding cytokine production after Ngo infection and supernatant treatment, the membrane array did not show a difference between the conditions, but ELISA analysis of cytokine induction does. It is essential to address this point in the discussion section.

This is not actually a difference in the results, but rather a nomenclature and labeling problem. The IL-17 in the membrane array is IL-17A, whereas our ELISA analyses are detecting IL-17C, which is a very different cytokine. IL-17A and IL-17C only share 27% identity. We checked with the manufacturer of the membrane array (Abcam), and they indicated that it specifically detects IL-17A, not IL-17C. We have updated the label on the figure in the supplement to specify that it is IL-17A.

Data are well described. However, in the methodology concerning RNAseq, it is essential to indicate which bioinformatics tools were used to model the results and make data comparisons. I mean, what statistical model was used to compare significantly different genes?

The information about the statistical model is found in the Materials and Methods section, as pasted in below. EBSeq is the empirical Bayesian model, and we provide the reference to Leng N et al. 2013. *Bioinformatics* 29:1035-43.

“Quantification of expression for each gene was calculated by RSEM, which produces both Transcripts Per Million reads (TPM) and expected read count(135). The expected read counts were used to determine differential gene expression using EBSeq(136). Average fold change was calculated for the set of four independent patient samples (average mock vs average *N. gonorrhoeae* treated, or average wild type treated vs average $\Delta ldcA$ treated). To be considered as differentially expressed, transcripts had to be $> 0.59 \log_2FC$ (1.5-fold) different for comparisons of wild type vs $\Delta ldcA$ or $> 1 \log_2FC$ (2-fold) different for comparisons of mock vs *N. gonorrhoeae*, with $P < 0.05$ and a false discovery rate (FDR) < 0.05 .”

Reviewer #3 (Remarks to the Author):

Using an unbiased approach, Garcia et al show that IL-17C drives inflammation mediated by *N. gonorrhoeae* or its culture supernatants in fallopian tube organ culture (FTOC) explants. They show expression of the IL-17C receptor chain, IL-17RE, on epithelial cells. They postulate that autocrine signaling by IL-17 through IL-17-RE mediates inflammation and fallopian tube damage. Inflammation could also be incited by peptidoglycan tripeptide monomers and also by treatment of fallopian tube explants with recombinant IL-17C. Detailed pathway analyses have been provided. The data are novel, clearly presented and support the conclusions. A couple of minor points need clarification.

1. Estrogen upregulates IL-17 expression by T cells. Is there any evidence that IL-17C or IL-17RE are estrogen (or progesterone) regulated? This would be an important consideration for inflammatory effects in the female genital tract.

We agree, and more investigation of the roles of estrogen and progesterone on gonococcal infections are warranted. In searching the literature, we found one observation with regard to IL-17C and hormones. Progesterone-treated, *Chlamydia trachomatis* infected endocervical cells showed increased IL17C transcript compared to estradiol-treated. We have included this information in lines 273-274, and referenced the study: Wan C et al. 2014. *Am J Reprod Immunol* 71:165-77. Given this information, we have made slight alterations to the statements in lines 50, 114, 272-273, 394-395, and 545.

2. Line 315 – I presume this is 200 ng/mL? How was this concentration selected? It seems to be much higher than the IL-17C levels detected in supernatants (eg, Fig. 3).

Yes, 200 ng/mL is correct. There is 1mL medium in the well to which 200ng is added. The range of concentrations we tried was based on the only report at the time that added recombinant human IL-17C to human cells: 4-500 ng/mL in Ramirez-Carrozzi V et al. 2011. *Nat. Immunol.*

12:1159. The concentration was clarified at the referenced line (now 331) and the above study cited at line 755 In the Materials and Methods.

REVIEWERS' COMMENTS

Reviewer #3 (Remarks to the Author):

The authors have addressed my critiques adequately and the paper is suitable for publication.

Reviewer #4 (Remarks to the Author):

After reviewing the manuscript resubmission, my opinion is that the authors have sufficiently addressed the points raised by Reviewer 2 and, where possible, have included new experiments that support their results.